# Self-Healable Lithium-Ion Batteries: A Review

**DOI:** 10.3390/nano12203656

**Published:** 2022-10-18

**Authors:** Ye Cheng, Chengrui Wang, Feiyu Kang, Yan-Bing He

**Affiliations:** 1Tsinghua Shenzhen International Graduate School, Tsinghua University, Shenzhen 518055, China; 2Laboratory of Advanced Materials, Department of Materials Science and Engineering, Tsinghua University, Beijing 100084, China

**Keywords:** self-healing, lithium-ion batteries, electrode, electrolyte

## Abstract

The inner constituents of lithium-ion batteries (LIBs) are easy to deform during charging and discharging processes, and the accumulation of these deformations would result in physical fractures, poor safety performances, and short lifespan of LIBs. Recent studies indicate that the introduction of self-healing (SH) materials into electrodes or electrolytes can bring about great enhancements in their mechanical strength, thus optimizing the cycle stability of the batteries. Due to the self-healing property of these special functional materials, the fractures/cracks generated during repeated cycles could be spontaneously cured. This review systematically summarizes the mechanisms of self-healing strategies and introduces the applications of SH materials in LIBs, especially from the aspects of electrodes and electrolytes. Finally, the challenges and the opportunities of the future research as well as the potential of applications are presented to promote the research of this field.

## 1. Introduction

With the widespread use of renewable and clean energy, the technology of energy storage has become one of the most practically significant research hotspots nowadays [1]. Among multiple kinds of energy storage devices, lithium-ion batteries (LIBs) have played an important role and dominated the portable electronics and electric vehicles (EVs) markets due to their massive advantages such as high energy density, reusability, and long service life [2]. Although LIBs have greatly improved our lives, many limitations still remain. Since the most widely used electrode materials are currently quite brittle and poorly scalable, they are susceptible to drastic structural changes and mechanical fractures in cycle processes, bringing about irreversible capacity loss and poor cycle stability [3]. To solve these problems, many efforts have been made to eliminate the cracks and pulverization in electrodes by optimizing the structural design and developing functional composite materials. Some recent research works have shown that introducing the concept of self-healing (SH) into battery materials can effectively enhance the stability and durability [4]. The LIBs containing self-healing materials and possessing self-healing features are defined as self-healable LIBs.

Self-healable LIBs consist of materials that can spontaneously heal the internal or external damages [5]. This concept takes inspiration from nature, since many animals such as starfish, gecko and even human skin can reconstruct injured parts and retain their original functions. The capability of the damaged part to repair itself is significant for the survival rates of these natural creatures. It is efficient to lengthen the cycle life and improve the stability and reliability of LIBs with the help of SH materials. Early research works concerning SH materials focused on encapsulation or hollow fibers that can release healing agents and polymerize to achieve a limited number of healings [6]. With further research, it was found that the multiple reversible healing can be obtained by physico-chemical approaches such as liquid metal and alloys, which can heal the cracks in electrodes via reversible solid–liquid conversion reaction [7]. In addition, the SH polymer binders have been carefully designed via dynamic bonds to stimulate spontaneous repair of the physical damages in electrodes. Furthermore, the highly stretchable solid-state SH electrolytes have been proven to effectively prolong the cycle life of solid-state LIBs.

To date, a variety of reviews describing recent advances in self-healing materials from different perspectives have been published [5,6,7,8,9,10,11,12,13,14,15]. This review, however, provides a systematical summary of the development as well as applications of SH materials in LIBs. Firstly, the healing mechanisms will be briefly described. Subsequently, we mainly focus on the principal problems with regard to the fading of LIBs properties and the corresponding SH strategies. The possible solutions of the internal failures in LIBs derived from the applications of SH materials in LIBs and the corresponding electrochemical performances are concluded, as shown in Figure 1. Finally, challenges and opportunities of the design for next-generation self-healable LIBs have been discussed and considered.

## 2. The Mechanisms of Self-Healing Materials

“SH” can be defined as the behavior of repairing itself and restoring functionality by using available resources in the event of damage [15]. The strategies associated with self-healing usually involve either physical or chemical events at the molecular level [5]. Depending on the differences in repair driving forces, SH mechanisms can be divided into physical approaches (i.e., phase-separated morphologies [21] and shape-memory recovery [22]), chemical approaches and physico-chemical approaches (i.e., encapsulation [23] and microvascular networks [24]). Since the vast majority of self-healing strategies applied to LIBs are chemical approaches, only this mechanism will be discussed in this chapter.

Chemical approaches are characterized by the reformation of reversible chemical bonds or supramolecular interactions. These approaches depend on inherent dynamic chemistry rather than introducing external repairing agents, so they are also known as the “intrinsic self-healing”. Due to the special repair mechanisms, there are two advantages to apply these approaches to LIBs: (1) Benefiting from the reversible bonding, the healing process is theoretically infinite; (2) Since no redundant electrochemical inactive components would be introduced to the battery system, the energy density of the battery could be optimized at the beginning. However, since the inherent dynamic chemical responses of the SH materials are usually slow, this self-healing mechanism dictates that the healing process may take longer.

Depending on the chemical bonding at the molecular level, the chemical approaches can be grouped into supramolecular chemistry as well as covalent and free-radical rebonding, which will be discussed respectively below.

### 2.1. Supramolecular Interactions

Supramolecular chemistry can reestablish 3D networks through dynamic formation of supramolecular bonds. The new “colloidal” fracture interfaces composed of several unbonded supramolecular bonds will get reformed once the network structure damaged, leading to the recombination of the interfaces and self-repair of damages [25]. Supramolecular bonds include various chemical interactions, such as hydrogen bonds [26]], metal-ligand coordination [27], host–guest interactions [28], ionic interactions [29] and π–π stacking [30]. For example, Qin et al. incorporated an ionic moiety into the π–π stacking system between naphthalene diimide (NDI) and pyrene (Py) derivatives and successfully tuned the self-healing temperature and Young’s modulus of the polymer [30]. After doping with Li(G4)TFSI, it displayed good ionic conductivity and could be used as a cathode binder in lithium–sulfur batteries.

Supramolecular rebonding processes are usually bottom-up and involve non-equilibrium states, the low bonding energy of these bonds (≈10 kcal mol^−1^) makes them strongly susceptible to external environmental influences and become easy to spontaneously reform due to their reversibility and sensitivity, which contributes to the reversibility of the process and their application in various room-temperature SH materials [31]. Another attractive feature of supramolecular chemistry is that the matrix can be repaired rapidly and autonomously, but the mechanical strength of the repaired sites is usually much lower compared to covalent and free-radical rebonding.

Supramolecular chemistry is favored in SH materials and complex battery systems due to the ease of dissociation–association and high energy efficiency. Among them, hydrogen bonds are widely used owing to their directionality and affinity. Hydrogen bonding refers to the dipole–dipole attraction between a hydrogen atom and an electro-negative atom with a lone pair of electrons (i.e., N, O, or F). In particular, hydrogen bonds based on urea-pyrimidine (UPy) groups are most widely utilized in SH materials because of their ease of application, good reversibility and strong temperature-mechanical property correlation (Figure 2a) [26].

Good SH properties are also achieved by metal–ligand (M–L) coordination interactions, which are formed between a coordination center and an organic ligand. The thermodynamic and kinetic parameters of M–L complexes can be tuned by coordinating different metal ions and ligand substitutes as well as by changing molecular parameters and ligand/metal ratios (Figure 2b) [32]. Compared to hydrogen bonding, M–L interactions are less sensitive to moisture, which is quite favorable for practical applications. In addition, host–guest interactions, ionic interactions and π–π stacking are promising candidates as SH materials. Ionic interactions are manifested by the formation of polymeric materials with ionic content less than 15 mol% and/or ionizable groups, which is called ionomer. π–π stacking interactions are facilitated by π orbitals where π-electron-rich molecules interact with π-electron-deficient groups. They have relatively few applications in LIBs and will not be discussed in detail here.

### 2.2. Covalent and Free-Radical Rebonding

Traditionally, the molecular synthesis of organic compounds is kinetically controlled, which will irreversibly form strong covalent bonds and single products [35]. During the last couple of years, researchers have found that, under equilibrium control by heating or irradiation, some covalent bonds can be reversibly formed, broken, or even reformed [36]. Since then, diverse methods have been exploited to design self-healing polymers with free-radical or covalent rebonding, which can be divided into three categories: condensation reactions, addition reactions and exchange reactions [11].

Condensation reaction is one of the most common chemical bonding methods in SH polymers, where two functional molecules react to construct a new bond and form a small molecule simultaneously. In most cases, this new molecule (usually water) is formed during bond formation and consumed in turn. For example, Sumerlin et al. successfully achieved bulk-state self-healing via boronic ester exchange at room temperature, which suggested that they may have promise for a variety of applications, including batteries (Figure 2c) [33].

Another typical reaction type is addition reaction, including the most widely investigated Diels–Alder (DA) reaction. This [4 + 2] cycloaddition reaction of a dienophile and a diene is controlled by dynamics and thermodynamics, making it particularly suitable for self-healing because temperature can be used as a convenient trigger. The bond between diene and dienophile will break when the bulk fractures and can be easily reconstituted through a simple procedure (Figure 2d) [34]. It is worth noting that no chemical by-products are produced in addition reactions.

The third category is the exchange reaction, where functional groups exchange at equilibrium state continuously and rapidly, making it easy to repair damages. Such exchange reactions include imines, oximes, siloxane, disulfide bonds and so on. Unlike reversible addition reactions, exchange reactions require lower stabilities of the corresponding reversible linker to achieve self-repair [37]. For example, disulfide bonds are much weaker, easier to control and able to induce self-heal at lower temperatures.

In general, SH materials can provide a suitable solution to the problems of LIBs due to their special mechanisms, such as electrode material deformation, poor interface contact, and capacity loss. We will focus on the possible applications of SH materials in LIBs in the following chapters.

## 3. Self-Healing Electrodes

The electrochemical performance decay of LIBs is usually caused by volume expansion/contraction during repeated charging/discharging processes due to the electrochemical reactions. Such a volume change will bring the formation of cracks, separate the active material from the collector and disrupt the electronically conductive network within the electrode. To overcome these problems, several research works have been done to improve the durability of electrodes, such as adding buffer materials, alloying the Li-active materials with inactive elements and building nanostructured electrodes. However, the addition of inactive or buffering components can reduce the gravimetric and volumetric energy densities of the battery. Nanostructured electrodes possessed low packing density and faced many other problems such as excessive consumption of electrolyte and repeated formation of the solid electrolyte interphase (SEI).

Recent studies have shown that SH materials can limit the formation of cracks/fractures on electrodes, increase the durability of electrode materials and thus enhance the cycle life of LIBs. One approach highlights the self-healing electrode material itself. Another approach is to coat the active material with a self-healing binder. Both of the approaches will be discussed successively in the following sections.

### 3.1. Liquid Metal and Alloys

Benefiting from fluidity and surface tension, liquid materials show great potential for self-healing. However, the batteries in previous studies usually need to operate at extremely high temperatures, which limits its application [38]. Therefore, gallium (Ga) draws much attention among potential candidates due to the fact that it is the only nontoxic metal with a low melting point at near room temperature (29.8 °C). This unique feature makes it a highly promising self-healing anode material, which possibly enables recoverable morphologies during repeated charging/discharging processes. Cracks caused by lithium insertion will disappear after deintercalation as a result of the ideal fluidity and surface tension of Ga. Such SH properties can potentially lead to an ultralong cycle life.

The self-healing behavior and electrochemical property of pure Ga was initially investigated. By forming the Li_2_Ga alloy, one Ga atom hosts two Li atoms when fully intercalated with lithium and provides a theoretical specific capacity of 769 mAh g^−1^ [39]. Saint et al. prepared single line phases Li_x_Ga_y_ by ball-milling and emulated the electrochemical reactivity of Ga towards Li. By in situ X-ray diffraction (XRD) measurements, they demonstrated that electrochemical-driven transformation from LiGa to Li_2_Ga was initially fully reversible, but adversely affects battery capacity retention during subsequent cycles. Eventually they stabilized the cell capacity at about 300 mAh g^−1^ beyond 20 cycles bypassing the structural transition [40]. Deshpande et al. performed electrochemical tests at 40 °C to ensure the liquid state of pure Ga [41]. Three intermetallic phases (Li_2_Ga_7_, LiGa, and Li_2_Ga) were formed. More importantly, they certificated the reversible solid–liquid transition upon lithiation and delithiation of Ga. The Ga electrode underwent crystallization upon lithium and transformed into solid state. Conversely, the solid-state Ga electrode would transform back to a liquid phase upon delithiation. Cracks formed in the solid state could be healed once the electrode returned to the liquid state. Therefore, it was easy to realize self-healing through the solid–liquid phase transition and improve electrode durability. In addition, the SH behavior of gallium nanodroplets (GaNDs) was demonstrated by Liang and co-workers via in-situ transmission electron microscopy (in-situ TEM) [42]. Similarly, those GaNDs encountered a liquid–solid phase transition and delivered SH properties.

However, there are still many issues that need to be addressed. Firstly, to ensure the liquid state of Ga, the working temperature of the aforementioned battery was usually higher than room temperature (RT). Secondly, the capacity of Ga was relatively low. Thirdly, Gallium nanoparticles (GaNPs) tend to aggregate as a result of their low surface energy and detach from the current collector upon cycling. Last but not least, although self-healing anode materials are not easy to pulverize upon cycling, the volume changes of Ga may cause cracks and other undesirable features. Therefore, the resulting anode lasted for only a few dozen cycles, and the cycling performance was not satisfactory. In the following chapter, we will summarize the research on Ga metal modification from these aspects.

#### 3.1.1. Working Temperature

High temperature is needed to ensure the liquid state of Ga for reasonable diffusivity. To decrease the melting point of Ga and improve capacity, Sn has been introduced to form Ga–Sn liquid metal (LM) alloy (Figure 3a) [43]. The melting point of the alloy was 20 °C, which is below the typical RT (25 °C). It was further stabilized in a skeleton formed by reduced graphene oxide together with a carbon nanotube. In situ microscopy was used to examine the self-healing ability of the LM alloy. As a result, the novel Ga-Sn alloy anode delivered an ultra-long cycle life (>4000 cycles with a capacity of about 400 mAh g^−1^ at 4 A g^−1^). Furthermore, Guo et al. constructed a liquid Ga-In metal system with a 15 °C eutectic melting point as well as a high capacity of 821.7 mAh g^−1^ [44]. The Ga-In LM anode in half-cells presented a steady capacity retention of over 99.98% per cycle after stabilizing. The superiority of the Ga-In LM anode system was further confirmed by full-cell tests via pairing with a LiFePO_4_ cathode. A high capacity, notable stability, and excellent rate capability was achieved. In addition, CuGaS_2_ hexagonal nanoplates have been successfully prepared via a vapor thermal transformation method by Song et al. [45]. The as-synthesized anode could operate in a wide temperature range and exhibited a stable capacity over 784 mAh g^−1^ at 0.5 A g^−1^ at the temperature of 318 K.

#### 3.1.2. Capacity

Among diverse anode materials, Si owns the highest theoretical capacity of 4200 mAh g^−1^ (lithiated to Li_4.4_Si). However, Si-based electrodes typically suffer from poor capacity retention due to the extreme volume change, which leads to material pulverization, loss of electrical contact with electrodes, continuous solid electrolyte interface (SEI) growth and consumption of electrolytes [49]. Based on this situation, Si can be the ideal reinforcement to make up for the lack of capacity of liquid metal and improve the energy density of the battery, At the same time, liquid metal can act as a liquid buffer and heal the fractures caused by the volume expansion/contraction of Si. Inspired by this strategy, Wu et al. mixed Si with Ga-Sn alloy [46]. The anode delivered a high capacity of about 670 mAh g^−1^ after 1000 cycles at 2A/g and a remarkable rate capability. Similarly, Han et al. reported a novel self-healing LM/Si nanocomposite material based on gallium-indium-tin (GaInSn) alloy as a smart anode for LIBs (Figure 3b) [16]. On the one hand, the spontaneous repairing LM nanoparticles absorbed the mechanical stress caused by the volume changes of Si. On the other hand, LM can act as the conducting media and ensure permanent electrical contact in Si electrode. The as-prepared nanocomposite anode obtained excellent electrochemical performances as characterized by a high initial coulombic efficiency (95.92%) and long-term stability (81.3% capacity retention after 1500 cycles at 8 A g^−1^).

In addition to the strategy of introducing Si, there are other modifications to the metal Ga that can also increase the capacity. Yarema and his co-workers reported a simple preparation method of colloid of Ga NPs with a capacity of 600 mAh g^−1^, which showed at least 50% higher reversible capacity than bulk Ga [50].

#### 3.1.3. Aggregation of Nanoparticles

Ga NPs are easy to get aggregated during charge/discharge processes owing to their low surface energy, so the cycling stability of electrodes based on monodispersed GaNPs was not satisfactory. To address this issue, the utilization of solid Ga-based compounds (e.g., GaS_x_, CuGa_2_, Ga_2_Se_3_, Ga_2_O_3_) seems to be the wise choice [51,52,53,54]. On the one hand, Ga-basd compounds could retain the SH properties of Ga metal upon the conversion reaction. On the other hand, it is easy to prepare various nanostructured anode materials, which can shorten the transport way and provide more active sites.

For example, GaNPs were replaced by superfine and highly dispersed Ga_2_O_3_ NPs by Tang and co-workers [55]. These Ga_2_O_3_ NPs were further embedded in carbon shells by a hydrothermal carbonization method to avert aggregation and enhance cycling stability. The resulting Ga_2_O_3_@C NPs anode showed a capacity of 721 mAh g^−1^ after 200 cycles at 0.5 A g^−1^, which was much better than previous studies based on Ga NPs. Recently, Guo et al. developed the hollow Ga_2_O_3_@nitrogen-doped carbon quantum dot nanospheres (H-Ga_2_O_3_@N-CQDs) as the anode material in LIBs (Figure 3c) [47]. Firstly, the anode can make use of the self-healing property endowed by Ga generated in the process of the conversion reaction. Secondly, the hollow core−shell framework helped relieve volume expansion, shorten transport paths, and provide more active sites. Benefiting from the aforementioned factors, the anode delivered a specific capacity of 700.5 mA h g^−1^ after 500 cycles at 0.5 A g^−1^. Furthermore, Ga_2_O_3_ nanoparticles with a size of about 4 nm were firstly obtained by a simple sol-gel method and then coated with different types of carbon materials. Specifically, reduced graphene oxide (rGO) offered more attachment sites for Ga ions and promoted the uniform distribution of Ga_2_O_3_ NPs. Consequently, Ga_2_O_3_/rGO anode exhibited higher capacity and better cycling stability (411 mA h g^−1^ after 600 cycles at a 1 A g^−1^) [56].

#### 3.1.4. Volume Changes

In addition, the most critical factor affecting the performances of LIBs is the volume expansion of the bulk materials. Carbon encapsulation is a well-known method of buffering volume changes. For example, Lee et al. prepared a liquid Ga electrode within a carbon substrate to minimize the detachment among solid-state LixGa particles [57]. In addition, the porous carbon matrix can suppress the volume changes of the embedded Ga. A recovery to initial liquid phase after multiple cycles was observed in the experiment sample, and the self-healing was not impressive without the confining matrix, which demonstrated the importance of a confining porous matrix. Moreover, core–shell fibers with encapsulated nanosized SH LM particles with well-designed void space were prepared by coaxial electrospinning and a carbonization process (Figure 3d) [48]. The free-standing anode offered a capacity of 552 mAh g^−1^ after 1500 cycles at 1 A g^−1^. The impressive electrochemical performance can be attributed to the unique core–shell structure and well-designed void spaces which can effectively alleviate the volume changes of the LMNPs during the lithiation/delithiation processes.

In general, it is an effective strategy to use Ga-based liquid materials as the self-healing electrodes in LIBs thanks to its low melting point, ideal fluidity, and comparatively high conductivity. To date, there are still many challenges to achieve the acceptable battery performances for practical application of Ga-based anode materials at room temperature. Numerous efforts should be devoted to further lowering operating temperature, inhibiting particle aggregation and improving cycle stability. It is apparent that this strategy is effective but limited for the improvement of battery performance. A simpler and more practical method is to prepare self-healing binders mentioned below.

### 3.2. Self-Healing Binders

Incorporating self-healing binders into electrodes, especially Si anodes, is another feasible way to enhance performance of LIBs. Most of the binders used in LIBs are polymers. Different from conventional polymer binders, SH polymer binders are stretchable and can spontaneously restore the cracks that arise from the volumetric changes of Si anodes. In this way, the mechanical and electrical connections among Si particles could be more stable, and better cycle stability could be achieved. According to self-healing mechanism, different types of SH binders based on supramolecular interactions as well as covalent and free-radical rebonding have been extensively researched. We will discuss each of these two different mechanisms of self-healing binders in following sections.

#### 3.2.1. Supramolecular Interactions

Supramolecular interactions include hydrogen bonding, metal–ligand coordination, host–guest interaction and dynamic ionic bonding. The vast majority of the SH polymer binders developed so far are based on supramolecular interactions. Although dynamic covalent bonds can be used to synthesis self-healing materials, supramolecular interactions are more favored in complex battery systems. This may be attributed to its energy-efficient feature (easy to dissociate-associate) and repeatable healing capability. This interaction can be formed between the binder as well as the active material and/or between the polymer chains.

Among different supramolecular interactions, hydrogen bonds are the most common. Hydrogen bonds can induce self-repair autonomously and repeatedly at room temperature. Polar functional groups on SH polymers could act as both hydrogen donors and acceptors [58]. Bridel et al. indicated that the strong hydrogen bonds were proposed to be critical for the electrochemical properties of Si-based anodes because of inducing the SH property [59]. This unique adaptable interaction can accommodate the dramatic volume changes of active materials and maintain the electronic wiring and integrity within the electrode.

Cui and co-workers were original pioneers who applied self-healing chemistry to silicon anodes to overcome their short lifespan (Figure 4a,b) [17]. It is worth noting that they used silicon microparticles (SiMPs) as electrode materials because, compared with silicon nanoparticles (SiNPs), SiMPs are more economical and available in the market. However, all previously reported batteries based on pure SiMPs anodes had an extremely short cycling life. Therefore, they prepared this amorphous self-healing binder with a modest conductivity and a low glass-transition temperature. Driven by the dynamic reassociation of hydrogen bonds, the anode showed mechanical and electrical self-healing ability as a result. The cycle life of SiMPs electrodes with an areal capacity of ≈1 mAh cm^−2^ were reported to be 90 cycles with a capacity retention of 80%, which was superior to conventional binders (i.e., PVDF = 14% and NaCMC = 27%). However, this still could not meet the commercial requirement of LIBs (3 mAh cm^−2^) or achieve long-cycle stability.

To achieve high mass loading and ideal areal capacity, Cui and his group realized 3D spatial distribution of a self-healing polymer (SHP) within the electrode together with Si particle size control (Figure 4c) [60]. Different from the initial work, where a thin layer of SHP was coated on the surface of Si particles, in this work, the anode was fabricated by repeatedly blading carbon black (CB) and SHP on Cu substrates under heating conditions. In this way, the SHP/CB composite would infiltrate into the whole thickness of Si anode to form a 3D distribution within the electrode. They also revealed that Si particle size control had a significant impact on the healing behavior and cycling performance. Compared to SiMPs with larger diameters (3.5 and 1.0 µm) and smaller diameters (30 nm), the optimal Si particle size (0.8 µm) complied with the requirements for low cost, high CE as well as good cycling performance (80% capacity retention after 500 cycles). Benefiting from the 3D distribution of SHP and carefully designed Si particle size, Si-SHP/CB electrodes showed good cycling stability at high areal capacity (3–4 mAh cm^−2^).

Then, in order to understand the effects of viscoelastic properties on battery performance, they prepared a range of supramolecular polymers with different crosslinking density [62]. The viscoelastic properties varied with the proportion of the trifunctional fatty acid. Binders with the relaxation time of around 0.1s delivered the best electrochemical performances because of the balance between viscoelasticity and mechanical strength. The large particle Si electrode (0.8 µm) with appropriate density of crosslinked junctions (from 43% triacid) could maintain 80% capacity after over 175 cycles. It was also found that binders were hard to realize self-healing under a high density of cross-linking.

Apart from high capacity as well as good cycling stability, the rate performance associated with fast charging and high-power density is also vital for practical Si anode implementation. Therefore, the polyethylene glycol (PEG) groups have been incorporated into the SHP to facilitate Li–ion conduction [63]. The SHP-PEG binder with high ionic conductivity could realize self-heal autonomously and spontaneously. Thanks to this strategy, the SiMPs anode could maintain electric conduction after multiple cycles, and show better rate performance than the previous SHP binders. When the ratio of SHP and PEG750 was controlled at 60:40 (mol%), the battery exhibited the best performance, including excellent rate performance, a high discharging capacity (about 2600 mAh g^−1^ at 0.5 C), and a long cycle stability (80% capacity retention after up to 150 cycles).

In addition, Hu et al. presented a new kind of conductive hydrogel binder with 3D continuous electron transport pathways for SiNPs anodes [64]. The three-dimensional hydrogel binder (named as ESVCA) was prepared by adding ammonium persulfate (APS) into the composite solution of PEDOT: PSS, PVA, and 4-carboxybenzaldehyde (CBA). APS functioned as the gelation agent to trigger the crosslinking polymerization of PEDOT: PSS and establish an interpenetrating network. PEDOT: PSS facilitated good electrical conductivity, while the massive hydroxyl groups of PVA were responsible for Li-ion conductivity and acted as hydrogen donors to form dynamic hydrogen bonds with the sulfonate groups of PSS. The Si-ESVCA electrode delivered a capacity of 1786 mAh g^−1^ at 0.5 A g^−1^ and 71.3% capacity retention after 200 cycles at ambient temperature. Recently, an SHP was designed and prepared by grafting poly(ether-thiourea) (TUEG) onto poly (acrylic acid) (PAA) via amidation reaction in the Si electrode drying process. Thanks to the zigzag hydrogen bonded array and the ether oxygen atom on TUEG, this binder (PAA-TUEG) can facilitate rapid Li ion conduction and possess a self-healing ability. To be specific, the Si electrode using PAA-TUEG binder achieved a Li-ion diffusion coefficient of 8.80 × 10^−5^ cm^2^ s^−1^ and outperformed the PAA binder in the field of rate performance. In addition, it exhibited a high initial coulombic efficiency (CE) of 87.2% and a capacity retention of 82% after 300 cycles. The electrochemical performance of LiFePO4//Si/PAA−TUEG full cell was also promising, which reflected the great potential of the binder for practical applications [65].

To further enhance the self-healing efficacy of hydrogen bonds, one method is to introduce the hyperbranched β-cyclodextrin (β-CDp) binder into Si anodes [66]. Different from one-dimensional material, β-CDp presented multidimensional hydrogen-bonding interactions due to the unique macrocycle structure. The enhanced Si-binder interaction could improve mechanical stability and solve the chronic insufficient life span of silicon anodes. Another method is to make use of the ureido-pyrimidinone (UPy) unit to form stable quadruple-hydrogen-bonding interactions [67,68]. For example, a small amount of UPy moieties were integrated with linear polymer poly (acrylic acid) (PAA) as a self-healing binder for the SiNPs anode (Figure 4d) [61]. Compared with previously reported self-healing binders, the supramolecular binder delivered higher healing efficiency due to the stable quadruple-hydrogen-bonded dimers. A high columbic efficiency (86.4%) and a long-term cycling stability (over 110 cycles) was achieved, which was superior to those Si anodes based on PAA, CMC, and PVDF binders. In addition, Yang et al. prepared ureido-pyrimidinone functionalized polyethylene glycol (UPy-PEG) as a self-healing binder that can realize self-healing without no external stimuli [69]. However, toxic organic solvents were required for the preparation of electrodes due to the poor solubility of the binder, which limited its practical application. Moreover, Kim et al. partially introduced UPy on PAA and further grafted PEG to this UPy-functionalized PAA and named this polymer binder PAU-g-PEG [70]. UPy was responsible for self-healing through H-bonding within molecules and PEG Boosted lithium-ion conductivity. Benefiting from these two functional groups, the Si electrode with the PAU-g-PEG binder retained a capacity of 1450 mAh g^−1^ after 350 cycles.

Recently, Chen and co-workers prepared a self-healing poly(ether-thioureas) polymer (SHPET) with balanced rigidity and softness for Si anode [71]. On the one hand, the hydrogen-bonding pairs helped to eliminate cracks and fractures within the electrode efficiently. On the other hand, the thiourea units were responsible for the appropriate mechanical rigidity and strong adhesion force. Consequently, the integrity of the Si@SHPET electrode was well maintained during volume contraction and impressive cycling stability was achieved (85.6% capacity retention after 250 cycles at 4.2 A g^−1^).

As discussed above, an SHP with abundant hydrogen bonds and appropriate viscoelasticity can greatly enhance the electrochemical performances of Si anodes. Not only hydrogen bonds, but some other supramolecular interactions such as metal–ligand (M–L) coordination can also impart self-healing properties to polymers. For example, inspired by mussel byssus cuticle, Jeong et al. reported a new kind of SH copolymer binder with an Fe^3+^-(tris)-catechol coordination bond (Figure 5a) [72]. The polymeric network based on M–L coordination exhibited a near-covalent elastic modulus and could easily reform due to the high rapport between metal and ligand. Therefore, the corresponding Si anode showed a capacity retention as good as 81.9% after 350 cycles.

In addition, host–guest interaction is also pursued in the binder design. For example, taking advantage of the host–guest interactions between hyperbranched β-cyclodextrin polymer (β-CDp) and a dendritic gallic acid, Kwon et al. achieved dynamic cross-linking in SiNPs anodes (Figure 5b) [73]. The unique dynamic cross-linking was homogeneous among polymer chains and introduced a spontaneous self-healing. The structural stability of Si film was well maintained during repeated cycles. The SiNPs electrode (mass loading 0.8 mg cm^−2^) achieved 90% capacity retention after 150 cycles.

In addition to those mentioned above, dynamic ionic bonding has been playing a positive role in self-healing materials since it involves two oppositely charged ions and is stronger than hydrogen bonding in general. Sottos and co-workers covalently attached amine groups to the surface of SiNPs (Figure 5c) [74]. Then, dynamic ionic bonds were formed between amine functionalized Si particles and the carboxyl groups on the poly (acrylic acid) (PAA) binder during electrode fabrication. Thanks to the dynamic ionic bonding, the as prepared anode exhibited an outstanding cycle stability with 80% retention after 400 cycles at 2.1 A g^−1^. In addition, by means of Meldrum’s acid-based binders, Kown et al. carried out a molecular-level systematic study to reveal the key parameter of the binder design for Si anodes [75]. Three different polymeric binders were investigated: binders with no/weak interactions (i.e., PVDF), with covalent crosslinking (i.e., K_100_) and with supramolecular interactions (C_100_). The cycling stability of LIBs varied with binders and the SiMPs anode with C_100_ showed the optimal performance (51% capacity retention after 500 cycles. This was attributed to the strong ion–dipole interactions and the induced self-healing properties.

#### 3.2.2. Covalent and Free-Radical Rebonding

Although self-healing polymers based on supramolecular interactions have been demonstrated as potential binders for Si anodes, there are still some issues that need to be solved. For example, the supramolecular network solely based on hydrogen bonds deforms easily, and the diffusivity of lithium ion is insufficient. To solve the above problems, boronic crosslinker (BC) was incorporated into the guar gum by Ryu et al. to generate boronic esters spontaneously art room temperature [76]. Guar gum were rich in hydroxyl groups and ensured good contact between the current collector and the silicon particles. BC reacted with vicinal alcohol without additional driving forces, while the other component, polyethylene oxide (PEO), contributed to the enhanced Li-ion conductivity. As a result, the boronic crosslinked guar (BC-g) binder extended the cycle life of either SiNPs anodes or SiMPs anodes. The electrode exhibited a good capacity retentivity and rate performance even under rigorous conditions (i.e., with high mass loading). Moreover, based on the reversible nature of Diels–Alder click chemistry, 1,6-bismaleimide (BMI) was added into furfurylamine-functionalized poly (acrylic acid) (FPAA) to form a 3D crosslinking polymer network, which was expected to be capable of self-healing [77]. The proposed Si electrode with the DA-PAA binder had a capacity of 1076 mAh g^−1^ after 200 cycles at 0.5 C.

#### 3.2.3. Dual Crosslinking

In addition to the physical mechanism based on supramolecular interactions and chemical mechanism based on dynamic covalent bonds mentioned above, dual crosslinking, through the complex structural crosslinking of the polymer itself and the abundance of hydrogen bonds, has become another mechanism that can also realize the polymer binder to achieve self-healing ability.

Xu et al. synthesized a water-soluble polymer binder poly (acrylic acid)-poly (2-hydroxyethyl acrylate-co-dopamine methacrylate) (PAA-P(HEA-co-DMA)) based on dual crosslinking [78]. This self-healing binder was modulated by soft and rigid domains. The self-healing property was attributed to the multiple networks structure and massive hydrogen bonds. It could withstand violent volume deformation of SiMPs anodes due to the unique dual crosslinked networks. The SiMPs anode had 93.8% capacity retention after 220 cycles at 1 A g^−1^. Furthermore, Jin et al. prepared a self-healable polyelectrolyte binder for anode materials which have drastic volume changes and poor electrical conductivity (i.e., silicon, sulfur) [79]. Poly(acrylamide-co-2-(Dimethylamino) ethyl acrylate) (poly (AM-co-DMAEA)) copolymers were synthesized via radical polymerization. Then, polydopamine and phytic acid were introduced to form a crosslinked polyelectrolyte binder network through reversible hydrogen bonds and ionic bonds. Notably, this process was in-situ and occurred spontaneously without any triggers. Benefiting from the dual-crosslinked networks, massive polar groups together with P=O group, the polyelectrolyte binder delivered self-healing property, good mechanical strength, and ionic conductivity.

To date, the self-healing binders involving supramolecular interactions and/or dynamic covalent bonds have received intense attention due to their reversible nature. SH binders based on supramolecular interactions, especially hydrogen bonding, are most attractive and have been extensively studied because they are easy to dissociate–associate and can achieve repeated self-healing at room temperature. Although the aforementioned binders are promising to improve the cycling stability of Si anodes as summarized in Table 1, there are some problems that remain to be solved.

Firstly, since binders do not contribute to lithium storage and conductivity, and excess binders will reduce the energy density of LIBs instead, the amount of binders should be carefully controlled and reduced as much as possible. Ideally, it should be less than 5 wt%, as in commercial graphite. However, the binder content in the literature reported above is typically 10% or even more. This is because reducing the binder content in silicon electrodes always brings poor performances due to the failure to provide sufficient mechanical stability and adhesion between electrode components. One method is to distribute carbon black (CB) in SHP to promote electrical conductivity and ensure electrical self-heal [60]. The drawback is that CB particles will slowly form aggregates in the polymeric network during repeated cycling, and the conductivity will reduce with time. Although we can enhance the CB–SHP interaction or prepare Si particles with conductive coatings, the problem has not been fundamentally solved. Another method is to incorporate ion-conducting and/or electron-conducting molecules into SHP [63,70]. In this way, the transport of lithium ions will be facilitated and the conductive network will be well maintained. The above studies investigated SHP with specific functions, but the effect on the overall cell performance is still unclear and needs further investigation. Furthermore, intrinsically conductive polymer binders are promising, but the enhancement of these properties in binders often comes at the expense of other properties such as mechanical and adhesion properties. Overall, a single polymer binder cannot meet all requirements, so a rational design incorporating multiple functions will be essential.

Secondly, despite considerable progress in the development of SHP, the commercialization of silicon anodes is still hampered by various challenges and unmet goals. Most of the binder systems reported previously have only been evaluated in half-cells at moderate temperature. In order to perform a rigorous evaluation of binders and its impact on cycle stability, electrochemical testing must be performed in full cells under harsh conditions. In addition, most of these binders are applied in SiNP anodes. SiMPs are more promising for practical applications since they are simple to prepare and commercially reproducible at low cost. In the future, investigations in SiMPs or Si/Ga composite anodes will be needed.

Thirdly, the potential of silicon anodes with self-healing binders in flexible lithium-ion batteries has not yet been fully explored. Flexible lithium-ion batteries are crucial for advancing various areas, such as portable and wearable electronics. To date, a range of self-healing polymers have been investigated to heal damage spontaneously. Although they can endure some little deformation, like bending or rolling, they tend to become fractured under complex deformations. In addition, it would be difficult to achieve efficient self-healing for all the components if the battery structure is not reasonably designed. In addition, once the LIB is damaged, the conventional organic electrolyte decomposes rapidly in air, which further increases the difficulty of self-healing. To resolve this problem, Zhao et al. successfully prepared an SH aqueous LIB with the anode and cathode respectively made up of LiTi_2_(PO_4_)_3_ (LTP) and LiMn_2_O_4_ (LMO) nanoparticles loaded into CNT sheets [80]. Even after repeated cutting and cycles, the electrochemical storage capacity was well maintained due to the prominent SH ability. Rao et al. reported a self-healable and flexible all-fiber-based quasi-solid-state LIB [81]. The porous reduced graphene oxide (rGO) fibers with SnO_2_ quantum dots and spring-shaped rGO fibers with LiCoO_2_ were made up as its anode and cathode, respectively. The battery showed great flexibility together with SH ability at room temperature and thus could withstand daily deformation and damages in wearable electronics. Hao et al. fabricated an omni-healable and tailorable aqueous LIB [82]. Benefiting from dynamic reversible covalent bonds, the device could quickly recover to configuration integrity and restore the mechanical and electrochemical properties once damaged. Cui and his co-workers synthesized stretchable graphitic carbon/Si anode with the help of conformal coating of a highly elastic SH polymer [83]. The composite anode presented a 719 mAh g^−1^ overall capacity and 81% capacity retention after 100 cycles. These works propose effective ways to develop high-capacity Si anodes for stretchable LIBs, and further research is needed to realize their practical application in flexible LIBs.

## 4. Self-Healing Electrolytes

Organic liquid electrolytes cause the major safety issues of commercial LIBs due to its flammability, danger of leakage as well as high sensitivity to water vapor. Replacing the liquid electrolytes with solid-state electrolytes is considered to be an effective solution to the above problems. Solid-state electrolytes could be grouped into two categories: gel polymer electrolytes (GPEs) and all-solid-state electrolytes (ASSEs). GPEs have both good safety and high ionic conductivity similar to organic liquid electrolytes. In addition, they are light, viscoelastic, easy to processs, and have low reactivity with the electrode material. However, due to the swelling of the polymer in the electrolyte, GPEs usually have poor mechanical properties and flammable nature, which still presents hidden danger. Moreover, the development of ASSEs, including polymers, ceramics and their hybrids, was given high expectations. Solid polymer electrolytes (SPEs) possess good flexibility as well as process ability, but the low ion conductivity (<10^−4^ S cm^−1^) and easy-to-fragment features limit their utilization. Based on these considerations, the strategy of modification of solid electrolytes containing polymers to give them self-healing capabilities has become a quite promising method to solve the unstable cycle problems of LIBs and to extend the life of batteries through spontaneous healing damages.

### 4.1. Self-Healing Gel Polymer Electrolytes

#### 4.1.1. Liquid Electrolytes

Flammable organic electrolytes are subject to leakage problems in extreme cases. In addition, as a result of the poor adhesion between the separator and the electrode, undesirable slips or gaps with high contact resistance may form at the separator/lithium metal interface during prolonged cycles, resulting in significantly non-uniform Li^+^ flux distribution. Subsequent growth of lithium dendrites and pulverization of lithium metal can be expected. Gel polymer electrolytes (GPEs), which combine traditional liquid electrolytes and solid polymer electrolytes, could largely improve the safety of LIBs and enhance the electrolyte/electrode interface contact.

For example, Liu and co-workers prepared a composite polymer electrolyte (CPE) membrane composed of fatty acid based self-healing polymer (SHP) and Li+-conducting nanoparticles (Ga-doped Li_7_La_3_Zr_2_O_12_, LLZGO) (Figure 6a) [84]. Then, it was soaked in liquid electrolytes to enhance the ionic conductivity. The as-prepared Li foil symmetric cells and Li|Li_4_Ti_5_O_12_ cells deliver superior cycling life, which could be attributed to the self-healing property, strong adhesion, and volume conformity of CPE (Figure 6b). Notably, a flexible, wearable as well as self-healable aqueous lithium-ion yarn battery was prepared by in-situ polymerization of calcium ion cross-linking sodium polyacrylate and sodium alginate (PANa-Ca-SA) [85]. The battery maintained 70% of its capacity even after 5000 cycles (Figure 6c). The extraordinary service life of the yarn battery contributed to the microscopic and macroscopic SH properties of the hydrogel.

#### 4.1.2. Ionic Liquids

Compared with conventional liquid electrolytes, ionic liquids (ILs) exhibit many favorable features, including high ionic conductivity, strong thermal stability, and flame retardancy [87]. ILs can also be incorporated into polymer networks to fabricate GPEs for LIBs, and the polymeric ionic liquids (PILs) also exhibit flame retardancy and superior processability [88]. For example, Watanabe and co-workers synthesized a micellar ion gel composed of IL and a diblock copolymer through multiple hydrogen bonds [89]. The formed ionic gels showed self-healing properties after 3 h in the absence of external stimuli at room temperature.

Guo et al. prepared a highly conductive GPE with the capability of self-healing by immobilizing a kind of IL into a hydrogen-bonded network of PIL copolymers bearing UPy pendant groups [90]. This as-fabricated ionogel delivered a high ionic conductivity of >10^−3^ S/cm and was also flexible and nonflammable, showing great promise for next-generation flexible LIBs. Then, strong adhesion between the ionogel and the electrode ensured conformal contact lowered the interfacial resistance and helped suppress the Li dendritic growth. Accordingly, the Li|ionogel|LiFePO_4_ battery delivered a capacity of 147.5 mAh g^−1^ after 120 cycles at 0.2 C.

In addition, based on fully zwitterionic polymer networks in ion-dense electrolytes containing Li cations and weak basic anions, namely solvate ionic liquids (SILs), D’Angelo et al. prepared a novel solvate ionogel electrolyte [86]. Benefiting from the coulombic interactions between zwitterionic moieties and the SIL cations via dynamic physical crosslinking, the as-prepared GPE showed self-healing behaviour and almost completely recovered the initial mechanical property (Figure 6d).

### 4.2. Self-Healing All-Solid-State Electrolytes

#### 4.2.1. Solid Polymer Electrolytes

Emerging technologies are driving the continuous development of all-solid-state electrolytes (ASSEs) in LIBs to enhance safety and room-temperature performance. A successful ASSE must possess high conductivity, keep good contact with electrodes and encompass a cost-effective, scalable processability. Different from inorganic solid electrolytes which suffer from the unstable and high resistance electrolyte/electrode interface, solid polymer electrolytes (SPEs) possess superior processability.

Given that some breakable SPEs may lead to catastrophic battery failure caused by a short circuit, Xue et al. have designed a series of self-healing SPEs via quadruple hydrogen bonding [91,92,93]. One of the formed SPEs was able to cure the cutting damage within 2 h at 30 °C with no external stimulus and provide high stretchability. However, the healing efficiency was not satisfactory, and it usually needed some stimulus such as high temperature or a period of time. To enhance the healing efficiency at mild temperature, they prepared a novel SPE carrying urea groups and disulfide bonds (Figure 7a,b) [94]. The disulfide bonds were introduced due to the low activation energy of a disulfide metathesis reaction. The dynamic dual crosslinked network formed by hydrogen as well as disulfide bond interactions endowed this SPE with outstanding self-healing ability at room temperature. The Li|3PEG-SSH|LFP battery presented an outstanding capacity retention of 97.5% after 100 cycles.

Single-ion conducting polymer electrolytes (SICPEs), where anions are covalently tethered to the polymer backbone and Li^+^ is the only mobile species, have drawn much attention due to their high lithium-ion transference number. Nevertheless, the majority of SICPEs exhibit low σ at room temperature. To address this issue, Ahmed et al. prepared solvent and plasticizer free PEA-LiFSI-based self-healing SICPE with high σ, t_Li_^+^, thermal stability together with flame retardancy [96]. The LiFePO_4_|PEALiFSI|graphite full cell exhibited outstanding long-term cycling performance (144 mAh g^−1^ at 0.1 C after 500 cycles with a capacity retention of 95.0%).

#### 4.2.2. Solid Composite Electrolytes

Whiteley et al. developed a solid electrolyte-in-polymer matrix to form the electrolyte layer in LIB (Figure 7c,d) [95]. The membranes were synthesized by the hot pressing of powders of Li_2_S–P_2_S_5_ inorganic electrolytes and polyimine. Taking advantage of the void space among solid-state electrolyte pellets, they constructed an in-situ derived polymer matrix through reversible cross-links without sacrificing good contact. In this way, the weight proportion of solid electrolyte reached 80% and the thickness of the separator was reduced to about 4 µm. The battery based on this membrane with FeS_2_ as the cathode cycled stably for more than 200 cycles.

## 5. Self-Healing Current Collectors

In addition to the electrodes and electrolytes, current collectors are indispensable elements in LIBs, but are easily overlooked. Although current collectors cannot provide energy support, their quality greatly affects the cycling stability of LIBs, since the cracks or fractures will lead to the failure of the battery directly. Conventional current collectors generally rely on strain-adaptive structures requiring complex processing, and thus have poor stretchability, low device packaging density, and robustness, which is unacceptable especially for flexible LIBs. Therefore, the development of self-healing current collectors is of great importance to enhance the safety and stability of LIBs. However, current research on SH materials in LIBs is mainly focused on self-healing electrodes, self-healing binders, and self-healing electrolytes, while relatively little research has been conducted on self-healing current collectors.

Lee and co-workers demonstrated a self-healing current collector consisting of nickel sheets, eutectic gallium indium particles (eGaInPs) as well as carboxylated polyurethane (CPU) (Figure 8a) [18]. It delivered an initial electronic conductivity of 2479 S cm^−1^ and high stretchability with 700% strain. Benefiting from the flow of liquid metal to the damaged sites together with mechanical healing through interfacial hydrogen bonding of the CPU matrix, the impaired areas can be electrically restored. When used in graphene nanosheet supercapacitors, the self-healing electrodes exhibited a long lifespan of 1000 stretching/releasing cycles and 600 charging/discharging cycles.

Recently, Wu et al. prepared a self-healing current collector, where EGaIn was embedded in a 3D-Cu foam (EGaIn@3D-Cu), to efficiently stabilize lithium deposition in the disrupted graphite anode (Figure 8b) [97]. 3D-Cu foam was responsible for providing more lithium nucleation sites and reducing the local current density due to its large specific surface area. In addition, liquid metal (EGaIn) could reestablish the electrical connectivity among the disrupted active materials. The graphite anode showed an initial capacity of 327.4 mAh g^−1^ at 0.4 A g^−1^ and a high capacity retention of 94.2% after 100 cycles with scratch damage.

## 6. Self-Healing Interfaces

In the previous chapters, we have discussed the application of SH materials as specific components in LIBs; however, the application of SH materials in LIBs is not limited to these. The charging and discharging processes of LIBs are dynamic processes, which are usually accompanied by internal reactions such as the decomposition of the electrolytes, the deposition and stripping of lithium-ions within anodes, and the interfacial reactions between electrodes and electrolytes, such as the construction of solid-state electrolyte interfaces (SEIs). The products of these reactions greatly affect the cycling stability of LIBs, and the properties of the SH materials can be used to improve the dynamic instability, especially for the electrode–electrolyte interfaces; for example, the integration of SH materials at the electrode/electrolyte interface contributes to improving interfacial properties as well as reducing side reactions from direct physical contact. In this chapter, we will discuss and summarize the SH materials that can stabilize the interface in the internal reactions of a battery from the dynamic perspective.

### 6.1. Self-Healing Electrolyte–Electrode Interfaces

Among inorganic solid-state electrolytes, Li_1.5_Al_0.5_Ge_1.5_P_3_O_12_ (LAGP) electrolytes draw a lot of attention thanks to the high ionic conductivity at room temperature as well as good air stability. However, there has been little progress in the successful application of LAGP in Li metal batteries due to the interfacial incompatibility between LAGP and lithium metal. It is assumed that the use of a self-healing polymer electrolyte (SHPE) as an interface in LAGP-based Li metal cells can maintain a stable and integrated electrode/electrolyte contact during cycling, thus eliminating Li|LAGP interfacial side reactions (Figure 9a) [19]. Therefore, Liu et al. prepared novel anolyte SHE (ASHE) and catholyte SHE (CSHE) based on the polymerization of a pentaerythritol tetraacrylate (PETEA) crosslinker and a 2-[3-(6-methyl-4-oxo-1,4-dihydropyrimidin-2-yl)-ureido] ethyl methacrylate (UPyMA) monomer in the presence of lithium salt and different electrolytes to meet the needs of anodes and electrodes. The flame-retardant SHEs ensured robust electrolyte/electrode interface contacts and homogeneous Li plating/stripping behaviors. Benefiting from the high ionic conductivity of LAGP pellet and SHEs (~10^−3^ S cm^−1^ at RT), the Li||LiMn_2_O_4_ (LMO) battery exhibited good cycling stability (80.3% capacity retention after 120 cycles).

### 6.2. Self-Healing Artificial Solid–Electrolyte Interfaces

The direct utilization of Li metal as anodes in LIBs is hampered by the instability of the Li/electrolyte interface, continuous dendrite growth, and drastic volume changes during cycling, resulting in rapid capacity degradation, internal short circuits, and even safety hazards [100]. Therefore, it is reasonable to prepare artificial solid–electrolyte interfaces (ASEIs) layers on lithium metal via in-situ or ex-situ methods, and these ASEIs usually have higher strength and thickness than natural SEIs. Even so, they still cannot meet the requirements because the mechanical stability of the in-situ formed ASEIs is usually poor, and the fabrication process of ex-situ ASEIs is tedious and stringent [101].

#### 6.2.1. Liquid Metal and Alloys

Inspired by Wu’s work [43], Zhang et al. considered that coating LM on Li metal was a promising way to achieve stable ASEI layers and dendrite-free Li metal [102]. LM could not only repair the cracks of SEI layers caused by drastic volume changes during long-term cycling, but also regulate uniform deposition of Li^+^, resulting in excellent full-cell cycling performance. When paired with LTO cathodes, the LMNP-Li|LTO full cell showed smaller polarization and better electrochemical performance. Moreover, gallium-indium (GaIn) alloys have been used as anodes to improve the capacity utilization and cycle durability of LIBs at room temperature due to their low melting point (<25 °C) [103]. For example, Han and co-workers prepared a self-organized composite electrode (Li/LixLMy) by simply blade-coating a 5-μm-thick LM (GaInSn) onto a polypropylene (PP) separator [104]. The LixLMy would spontaneously form after discharging a cell consisting of LM/PP and Li foil. It seemed a wise choice to use GaIn alloys to promote uniform and dendrite-free deposition of Li metal. However, GaIn LM usually has large particle size and specific surface energy, which hinders its uniform dispersion and can cause problems such as volume changes and structure crushing upon cycling. Therefore, Gong et al. decorated GaIn NPs on the surface of porous carbon layers to homogenize the Li^+^ flux [98]. An extended lifespan of over 900h was achieved in a symmetric cell at 1.0 mA cm^−2^ for 2.0 mAh cm^−2^ (Figure 9b).

#### 6.2.2. Supramolecular Interactions

Wang et al. prepared SHPs based on poly (ethylene oxide) (PEO) segments and UPy groups as robust and strongly adherent ASEIs for Li metal to achieve long-term Li plating/peeling cycles [99]. This supramolecular polymer (PEO-UPy) can be further stabilized after reacting with Li and in-situ forming an LiPEO–UPy coating layer. Similarly, self-healing capabilities were guaranteed by the UPy quadruple hydrogen bonds that triggered the intrinsic restoration of this ASEI. The strong adhesion between Li and the coating together with the homogenization of the fast Li^+^ flux prevented uncontrolled nucleation and inhibited dendritic growth. As a result, the electrochemical performances of symmetric cells and full-cells based on this LiPEO–UPy-coated Li (LiPEO–UPy@Li) were both improved (Figure 9c).

Recently, a poly (ethylene imine) (PEI) interlayer, cross-linked by imine bonding, was introduced on Li anodes to endow the SEI layer with self-healing capabilities [105]. Together with the trifluorophenyl moieties which can coordinate with Li^+^, the Li-PEI-3F anode showed good cycling stability (600 h in the symmetric Li||Li cells at 1.0 mAh cm^−2^). In addition, by dynamically crosslinking poly(dimethylsiloxane) (SS-PDMS) and adding SiO_2_ NPs as reinforcement fillers, Chang et al. prepared self-healing single-ion-conductive ASEIs [106]. The optimal ASEI film (containing 7 wt% SiO_2_) achieved a combination of outstanding performances, including excellent self-healing efficiency (96%), high ionic conductivity and lithium-ion transfer number (0.71). By coating such an ASEI on the Li anode, the symmetric cells delivered excellent performances (1340 h at 0.5 mA cm^−2^, 1.0 mAh cm^−2^).

## 7. Challenges and Opportunities

### 7.1. Challenges

So far, a large number of studies have demonstrated that imparting SH materials into a specific component of the battery can have considerable influences on the whole battery system. Microcapsule chemistry has successfully promoted battery durability via delivering and releasing desirable additives upon external triggers [107,108,109]. However, pure extrinsic self-healing in LIBs has hardly been reported, possibly for the sake of volumetric and gravimetric energy density. Conversely, intrinsic SH materials, including the design and manufacture of SH electrodes and electrolytes, are effective strategies for the treatment of mechanical damages in LIBs. Compared with other self-healing materials, intrinsic self-healing materials have advantages in volumetric and gravimetric energy density, but the slow response and self-healing process limit their commercialization. How to improve the self-healing efficiency is a major challenge in future research.

For liquid metal and alloys, Ga NPs are prone to aggregate due to the low surface energy and the electrode based on monodispersed Ga NPs presents poor cyclability, which impedes its practical applications. The utilization of solid Ga-based compounds seems to be a reasonable resolution, but the complex preparation process and reduced energy density of the battery are the issues that cannot be ignored. Therefore, reducing the production cost and simplifying the production process are the practical optimization methods. For self-healing binders, the integration of SH strategies into Si anodes has showed tremendous potential, but it is only in the circumstances where the content of the binders was very high (mostly >10%). In addition, most of the systems reported previously have only been evaluated in half-cells at moderate temperature, and there are few reports about self-healing full cells, so further exploration is needed. In general, there are still many aspects to be further explored and improved in the research of SH materials applied to LIBs.

### 7.2. Opportunities

In recent years, solid-state lithium-ion batteries have attracted increasing attention owing to their outstanding safety and high energy density. Compared to traditional liquid LIBs, solid-state LIBs use solid-state electrolytes (SSEs) and lithium metal anode instead of organic solvent electrolytes (OSEs) and graphite anode, which can easily reach a high energy density of 300 Wh kg^−1^ and avoid the risks of fire and explosion caused by electrolyte leakage, which is considered as the most promising alternative for the next generation lithium-ion batteries. However, as the most crucial part of the solid-state LIBs, SSEs are facing many problems, such as excessive thickness and poor interfacial contact. The thickness of current SSEs is typically more than 100 μm, which not only increases the overall impedance, but also reduces the utilization of active materials. However, the reduction in thickness will result in a sacrifice of mechanical strength, which may lead to the puncture of SSEs due to the growth of lithium dendrites on lithium metal anodes, causing a short circuit in the battery. Therefore, how to balance low thickness and good stability is a hot topic for SSE research [110].

Self-healing materials can effectively improve the cycling stability of batteries due to their unique self-healing capability, so this strategy can also be applied to the design of ultra-thin SSEs to help them complete self-healing when physical damage happens, which provides a feasible idea to solve this problem. In addition to effectively reducing the thickness, a self-healing strategy can also play a positive role in improving the interfacial contact between SSEs and electrodes. The integration of SH materials at the electrode/electrolyte interface contributes to improving interfacial properties as well as reducing side reactions from direct physical contact. In addition, the introduction of SH polymers in solid-state electrolytes inhibits dendrite formation and helps to restore ionic conductivity [111].

In recent years, many researchers have concentrated on the application of self-healing materials in SSEs. Guo et al. produced a highly conductive self-healable SPE containing both rigid and flexible chains [112]. Amino-terminated poly (ethylene glycol) (NH2-PEG-NH2) delivered dynamic hydrogen bonds which can be used as the supramolecular backbone. The corresponding all-solid-state LMBs showed excellent cycling stability and freely bending flexibility. Benefiting from the UPy dimers, Zhou et al. synthesized cyclophosphazene-based SH polymer electrolytes via photopolymerization [20]. It showed high thermal stability (>300 °C) and good SH capability, leading to improved safety performance of LMBs.

To sum up, we consider the significant potential of a self-healing strategy to optimize the reliability of solid-state batteries and the promise of SSEs for the future applications, and believe that the research combining SH materials with SSEs is well worth exploring.

## 8. Conclusions

Depending on the differences of healing mechanisms, self-healing behaviors can be categorized into physical approaches, chemical and physico-chemical approaches. The SH materials used in LIBs currently are mainly based on chemical approaches which can be grouped into supramolecular reactions and covalent/free radical rebonding. These two strategies achieve self-healing by means of either supramolecular interactions or reversible chemical bonds, which can achieve theoretically permanently healing but have relatively lower efficiency. Due to the deformation and fractures of the materials in LIBs, the capacity of the active materials would decrease in repeated charging/discharging processes and the battery lifespan become shortened. Fortunately, self-healing strategies provide new ideas to solve the above problems by healing the damaged parts to enhance the performance.

To date, a large number of LIBs with self-healing properties have been constructed. Liquid metals and alloys as well as self-healing binders are instrumental in reducing mechanical cracks/fractures on electrodes, especially for Si based anodes, which can help increase the durability of the electrode materials. In addition, SH materials have also been used to improve the mechanical properties of solid polymer electrolytes through spontaneous repair of the damages. Therefore, the interfacial properties, dendrite suppression as well as mechanical flexibility can be improved. In addition to the internal components, self-healing materials can also be applied to the current collectors. Furthermore, self-healing strategy can also be used to solve the problem of interfacial stability due to the generated by-products during cycling processes, such as self-healing artificial SEI layers. Self-healable LIBs exhibit longer durability, higher safety, and better performance, and the developing of LIBs based on self-healing materials is one of the most promising directions for their practical application. Although numerous efforts have been made in this area, many challenges still exist, such as further improvements of healing efficiency. Furthermore, the integration of SH materials in solid-state batteries can inhibit dendrite growth of a lithium metal anode and provide a feasible pathway for a high energy density battery. Overall, further understanding and development of SH mechanisms and materials are necessary especially for the accelerated development of self-healable LIBs either in the field of basic research or industrialization.

## Figures and Tables

**Figure 1 nanomaterials-12-03656-f001:**
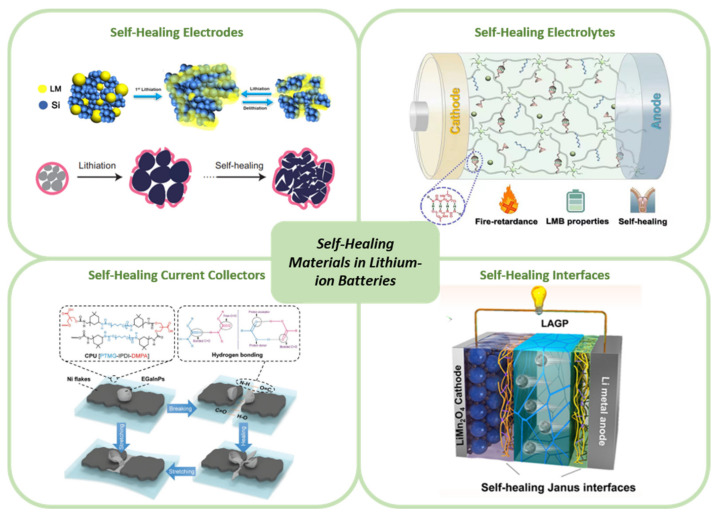
Self-healing materials in lithium-ion batteries. Images reproduced with permission as follows: Schemes of a charge–discharge process of the LM/Si anode [16]. Copyright 2018, Springer Nature, the comparison between a conventional silicon anode and a stretchable self-healing silicon anode [17]. Copyright 2013, Springer Nature. Schematics of the mechanism for the self-healing property of the Ni flakes–EGaInPs–CPU conductor [18]. Copyright 2018, John Wiley and Sons. Schematic illustrations of Li|LAGP|LMO batteries without interface layers and with GPEs and SHEs [19]. Copyright 2020, American Chemical Society. Schematic illustrations of flexible, self-Healing, and fire-resistant polymer electrolytes fabricated via photopolymerization [20], Copyright 2020, American Chemical Society.

**Figure 2 nanomaterials-12-03656-f002:**
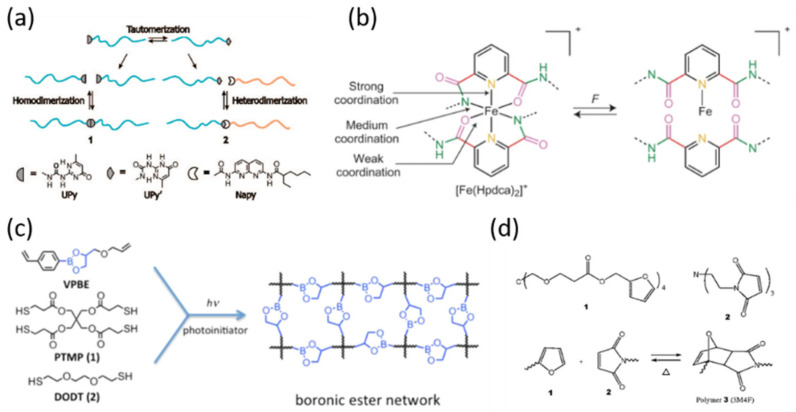
Typical self-healing mechanisms based on chemical approaches. (**a**) UPy groups in end-functional MHB polymers based on hydrogen bonds [26]. Copyright 2008, American Chemical Society; (**b**) the reversible rupture and reconstruction process of [Fe(Hpdca)2]+ based on M–L coordination interactions [32]. Copyright 2016, Springer Nature; (**c**) the synthesis of boronic ester network materials based on dynamic covalent bonds [33]. Copyright 2015, American Chemical Society; (**d**) reversible Diels—Alder reactions based on covalent bonds [34]. Copyright 2002, American Association for the Advancement of Science.

**Figure 3 nanomaterials-12-03656-f003:**
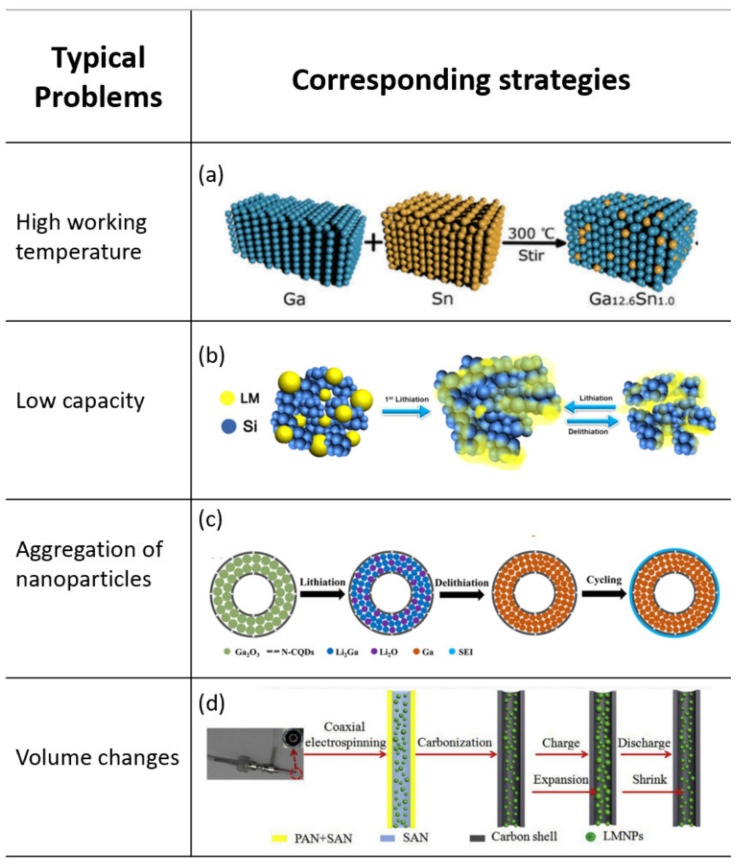
Typical problems and strategies for Ga-based self-healing electrodes. (**a**) synthesis procedure for the RGO/CNT-supported LMNP anode [43]. Copyright 2017, Royal Society of Chemistry; (**b**) schemes of charge–discharge process of the LM/Si anode [46]. Copyright 2018, Springer Nature; (**c**) flowchart of the possible structure evolution for H-Ga2O3@N-CQD nanospheres [47]. Copyright 2020, American Chemical Society; (**d**) the carbonization steps and the electrochemical processes for the LMNPs@CS fibers [48]. Copyright 2019, Elsevier Ltd.

**Figure 4 nanomaterials-12-03656-f004:**
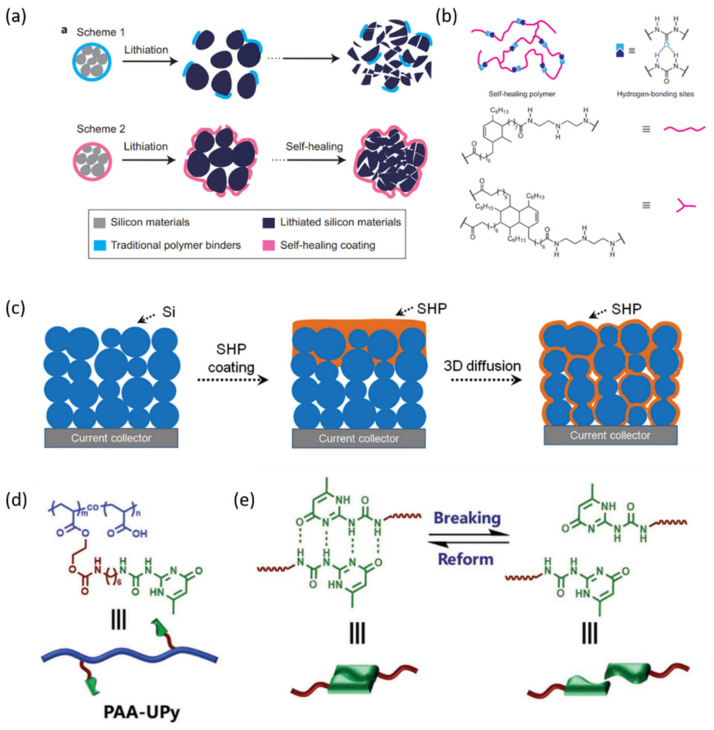
Binders based on hydrogen bonding for Si-based anodes in LIBs. (**a**) The comparison between conventional silicon anode and stretchable self-healing silicon anode and (**b**) molecular structure of the SHP based on hydrogen bonds [17]. Copyright 2013, Springer Nature. (**c**) schematic design of Si-SHP/CB electrodes [60]. Copyright 2015, John Wiley and Sons; (**d**) chemical structure of PAA–UPy supramolecular polymer and (**e**) UPy–UPy dimers could reversibly break and reform based on quadruple hydrogen bonding [61]. Copyright 2018, John Wiley and Sons.

**Figure 5 nanomaterials-12-03656-f005:**
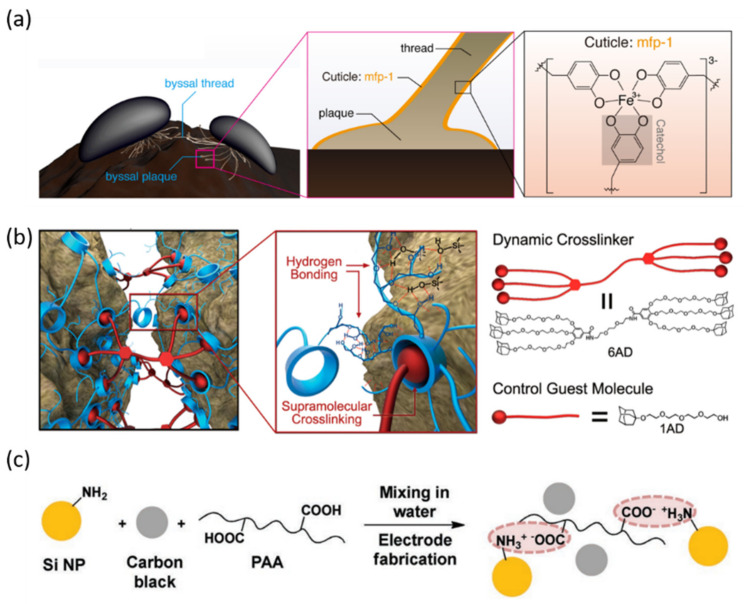
Binders based on other types of dynamic non-covalent bonds for Si-based anodes in LIBs. (**a**) mussel byssal thread with wet adhesion via Fe^3+^-catechol complex [72]. Copyright 2019, American Chemical Society; (**b**) proposed working mechanism of dynamic cross-linking based on host–guest interactions [73]. Copyright 2015, American Chemical Society; (**c**) the formation of interfacial ionic bonds between Si NPs and PAA binder [74]. Copyright 2017, John Wiley and Sons.

**Figure 6 nanomaterials-12-03656-f006:**
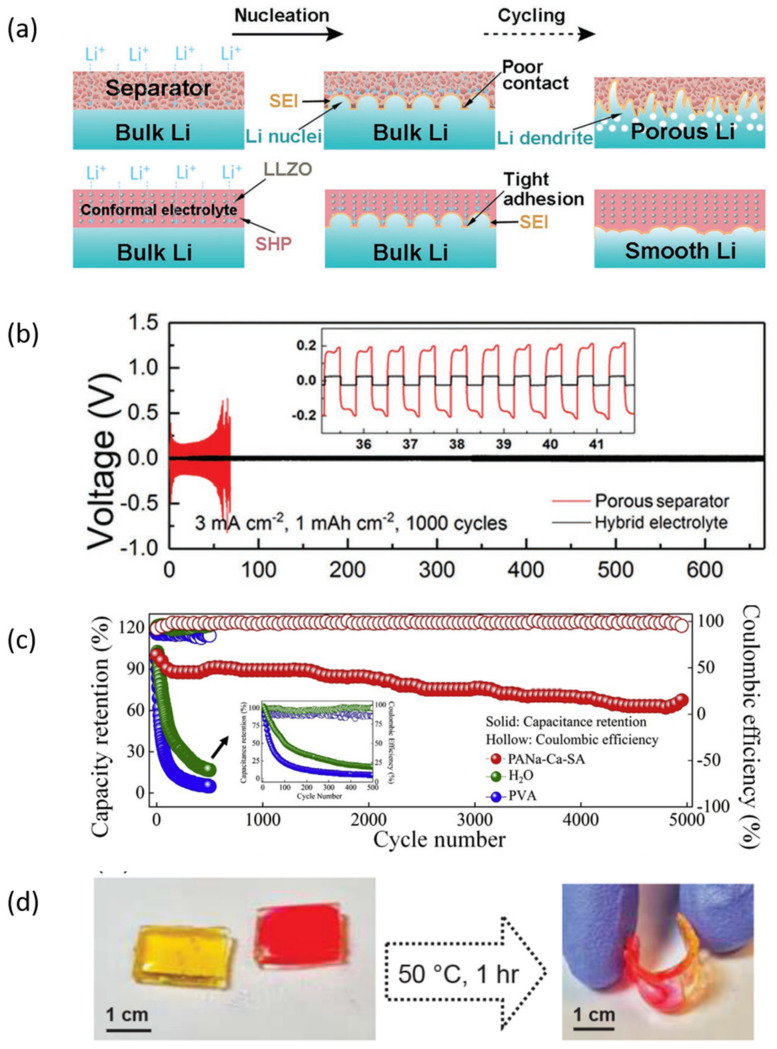
Self-healing gel polymer electrolytes in LIBs. (**a**) schematics of Li stripping/plating situation with porous separator and hybrid electrolyte, and (**b**) voltage–time profiles of the symmetric cell over long cycles using porous separator and hybrid electrolyte [84]. Copyright 2019, John Wiley and Sons; (**c**) the long cycle performances of ALIYB vs. PVA electrolyte and aqueous solution [85]. Copyright 2020, Elsevier Ltd; (**d**) photographs showing the self-healing of a rectangular bar-shaped gel initially cut into two pieces, each dyed a different color, and then heated at 50 °C for 1 h [86]. Copyright 2019, American Chemical Society.

**Figure 7 nanomaterials-12-03656-f007:**
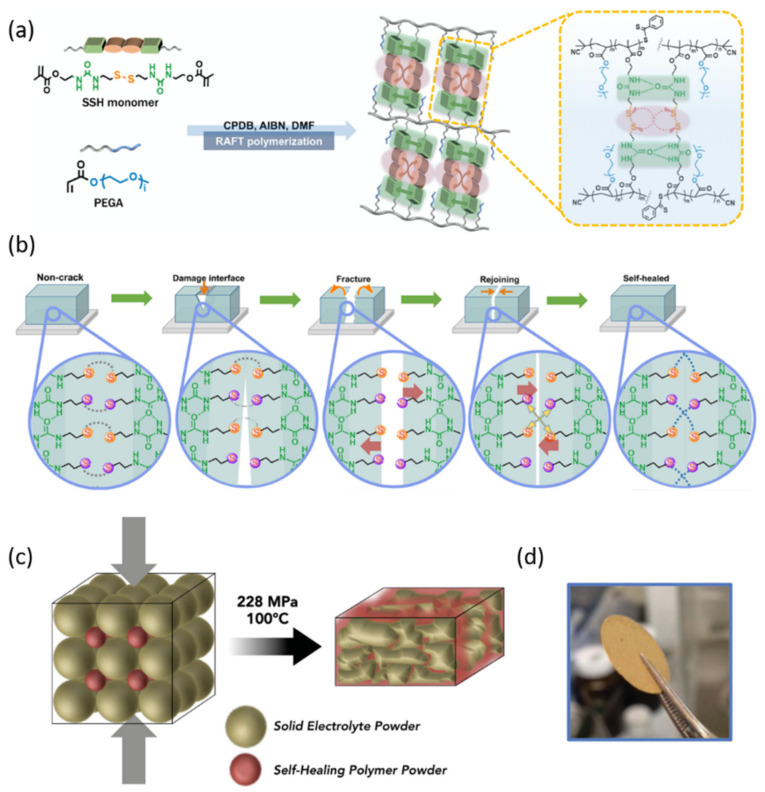
Self-healing strategies exploited in the electrolytes of LIBs. (**a**) schematic of the formation of PEG-SSH and (**b**) schematic of self-healing process with disulfide bonds and hydrogen bonds [94]. Copyright 2020, American Chemical Society; (**c**) schematic diagram of forming a solid electrolyte in polymer matrix membrane and a (**d**) picture of a free-standing SEPM separator 100 µm in thickness [95]. Copyright 2019, John Wiley and Sons.

**Figure 8 nanomaterials-12-03656-f008:**
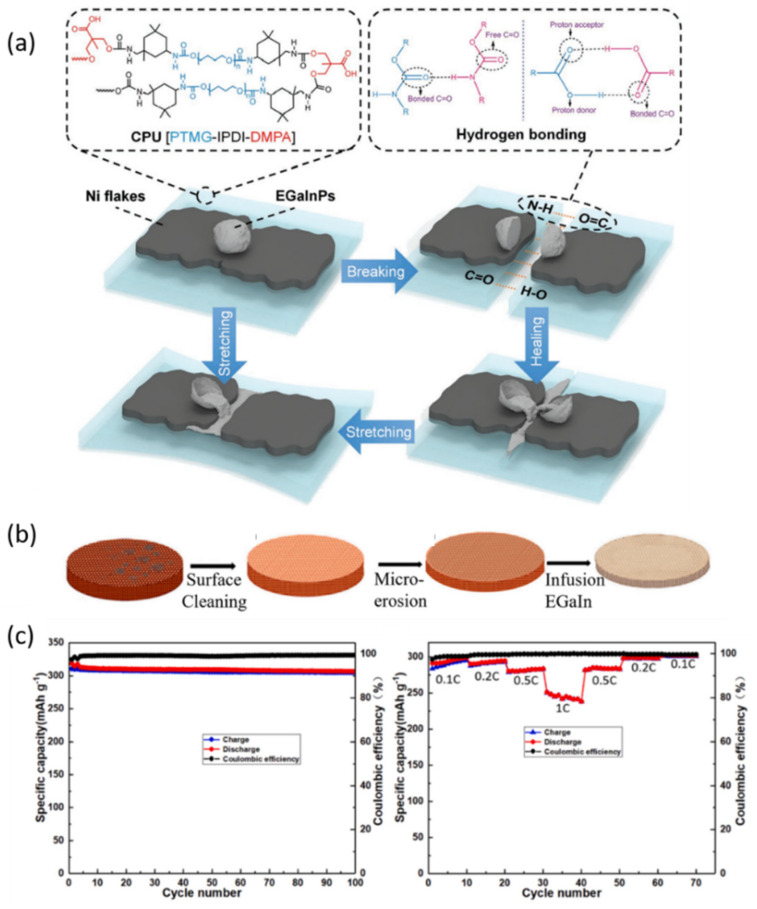
(**a**) Schematics of the mechanism for the self-healing property of the Ni flakes–EGaInPs–CPU conductor [18]. Copyright 2018, John Wiley and Sons. (**b**) schematic of the fabrication process of EGaIn@3D-Cu current collector and (**c**) cycling and rate performances of the self-healing graphite anode with scratch damage at 0.2 C [97]. Copyright 2021, Elsevier Ltd.

**Figure 9 nanomaterials-12-03656-f009:**
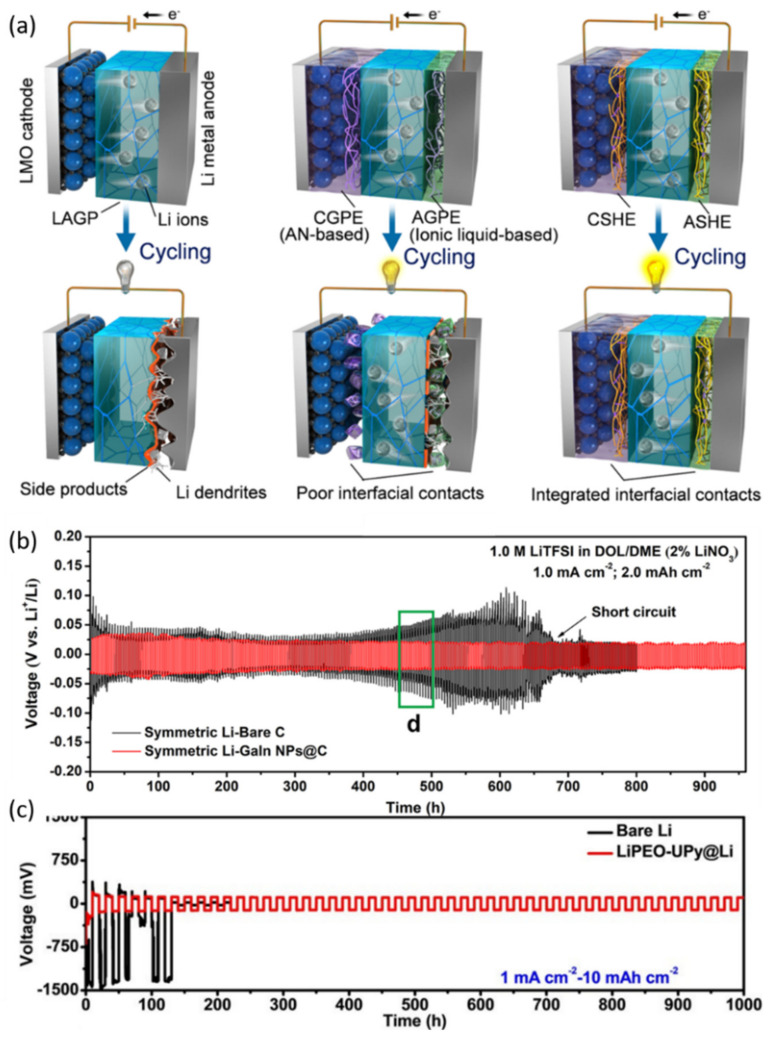
(**a**) Schematic illustrations of Li|LAGP|LMO batteries without interface layers and with GPEs and SHEs [19]. Copyright 2020, American Chemical Society; (**b**) voltage−time profiles of Li plating/stripping in Li-Bare C and Li-GaIn NPs@C symmetric cells at 1.0 mA cm^−2^ for 2.0 mAh cm^−2^ [98]. Copyright 2021, American Chemical Society; (**c**) voltage profiles of bare Li and LiPEO–UPy@Li anodes in symmetric cells at 1 mA cm^−2^ under 10 mAh cm^−2^ [99]. Copyright 2020, John Wiley and Sons.

**Table 1 nanomaterials-12-03656-t001:** Summary of the self-healing binders used in Si anodes reviewed herein.

Binder	Anode Mass Loading (mg cm^−2^)	Self-Healing Strategy	Specific Capacity (mAh g^−1^)	Capacity Retention	Ref.
SHP	SiMPs (0.5–0.7)	H-bond	2617	80% after 90 cycles at 0.4 A g^−1^	[17]
3D SHP	SiMPs (0.5–0.6)	H-bond	1100	~80% after 500 cycles at C/20	[60]
SHP-PEG	SiMPs (0.5–0.7)	H-bond	~2600 at 0.17 A g^−1^	80% after ~150 cycles at 1.7 A g^−1^	[63]
ESVCA	SiNPs (0.53)	H-bond	1786	71.3% after 200 cycles at 0.5 A g^−1^	[64]
β-CDp	SiNPs (0.3)	H-bond	2142	68.7% after 200cycles at 4.2 A g^−1^	[66]
PAA-UPy	SiNPs (0.4–0.6)	H-bond	4194	62.89% after 110 cycles at 0.8 A g^−1^	[61]
SHPET	SiNPs (1,2)	H-bond	3744 at 0.42 A g^−1^	85.6% after 250 cycles At 4.2 A g^−1^	[71]
Fe-PDBP	SiNPs (0.7)	Metal−ligand coordination	~1400	81.9% after 350 cycles at 1.5 A g^−1^	[72]
β-CDp/AD	SiNPs (0.8)	Host-guest interactions	~1600	90% after 150 cycles at 1.5 A g^−1^	[73]
PAA	SiNPs (0.5–0.8)	Dynamic ionic bonds	1150 at 4.2 A g^−1^	~80% after 400 cycles at 2.1 A g^−1^	[74]
C_100_	Si (0.2)	Dynamic ionic bonds	~2700	51% after 500 cycles at 3 A g^−1^	[75]
BCx-g	SiNPs (0.7)	Boronic-ester linkage	>2750 at 0.05 C	87.3% after 100 cycles at 0.2 C	[76]
DA-PAA	SiNPs (0.5–0.6)	Diels–Alder chemistry	2607	41.3% after 200 cycles at 0.5 C	[77]
PAA–P(HEA-co-DMA)	SiMPs (~1)	Dual crosslinking	~2550	93.8% after 220 cycles at 1 A g^−1^	[78]

## Data Availability

Not applicable.

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
