# Peer review of "Self-Healable Lithium-Ion Batteries: A Review"

_nanomaterials, 2022, doi:10.3390/nano12203656_

Round 1
Reviewer 1 Report (New Reviewer)
The authors provided a review about self-healing technology in Lithium-Ion batteries. The work summarizes current knowledge and points out challenges and opportunities.
1) Writing, especially in sections 1 and 2, must be improved. It is recommended to use a native/professional English speaker for proofreading and correcting the grammar and sentences.
2) When the references are related to the specific sentence, please insert them before full stop (i.e., “[x].”) and not after. You can see the authors’ instructions at https://www.mdpi.com/journal/nanomaterials/instructions.
3) It is recommended to invert the text in the bottom plane of Figure 1, so it can be comfortably read.
Author Response
Referee 1:
Comments to the Author:
The authors provided a review about self-healing technology in Lithium-Ion batteries. The work summarizes current knowledge and points out challenges and opportunities.
(1) Writing, especially in sections 1 and 2, must be improved. It is recommended to use a native/professional English speaker for proofreading and correcting the grammar and sentences.
(2) When the references are related to the specific sentence, please insert them before full stop (i.e., “[x].”) and not after. You can see the authors’ instructions at https://www.mdpi.com/journal/nanomaterials/instructions.
(3) It is recommended to invert the text in the bottom plane of Figure 1, so it can be comfortably read.
Reply: Thank you very much for your positive comments. We have carefully revised the manuscript according to your valuable suggestions.
- Writing, especially in sections 1 and 2, must be improved. It is recommended to use a native/professional English speaker for proofreading and correcting the grammar and sentences.
Reply: Thank you very much for your comment. We have carefully revised Chapter 1 and rewritten Chapter 2. In Chapter 1, we have simplified some phrases into words (e.g., replace “occupied a dominant position” with “dominated” on page 1, line 26), and used more authentic English expressions (e.g., replace “which presents” with “they are susceptible to” on page 1, line 30). We also correct the grammar and sentences (e.g., replace “renewable clean energy” with “renewable and clean energy” on page 1, line 23).
Due to the adjustment of the content and structure of the manuscript, we have rewritten Chapter 2 (“The mechanisms of self-healing materials”, from page 2 to page 5) to make it more concise in terms of language and content. This is because we realized that some kinds of self-healing materials are not used in lithium-ion batteries, and their mechanisms also have no relationship with lithium-ion batteries. In order to highlight the main point of this article, we have reclassified the self-healing mechanisms by deleting the classification of extrinsic and intrinsic methods and unifying them into physical, chemical and physico-chemical approaches. Self-healing materials used in lithium-ion batteries are basically based on chemical approaches, and other approaches are barely used, so we only discussed about chemical approaches.
We have also re-edited figure 1 (on page 2) and figure 2 (on page 4) respectively. All the revisions are in “Track Changes” mode and have been high lightened in the manuscript.
- When the references are related to the specific sentence,please insert them before full stop (i.e., “[x].”) and not after. You can see the authors’ instructions at https://www.mdpi.com/journal/nanomaterials/instructions.
Reply: Thank you very much for your suggestion. As you suggested, we have corrected the position of references throughout the text and placed them before full stop. All the revisions are in “Track Changes” mode and have been high lightened in the manuscript.
- It is recommended to invert the text in the bottom plane ofFigure 1, so it can be comfortably read.
Reply: Thank you very much for your suggestion. We have replaced the old Figure 1 with a new schematic (on page 2) which is more concise and explicit. All the revisions are in “Track Changes” mode and have been high lightened in the manuscript.

Reviewer 2 Report (New Reviewer)
The cycling of the battery is a dynamic process of the electrode material, which include material expansion and contraction, electrode crack and swelling, etc. Maintaining a good physical condition of the electrode is another important thing to extend the lifetime of Li-ion battery.
Here this review work systematically summarizes the self-healing materials used in battery, which is interesting and might attract lots of attention to this hot topic. The reviewer supports the publication. Though this review is a comprehensive document, if some more discussion could go to the disadvantages of the self-healing materials, would be better. The future is bright, while the difficulties to achieve should also make it clear.
Author Response
Referee: 2
Comments to the Author:
The cycling of the battery is a dynamic process of the electrode material, which include material expansion and contraction, electrode crack and swelling, etc. Maintaining a good physical condition of the electrode is another important thing to extend the lifetime of Li-ion battery.
Here this review work systematically summarizes the self-healing materials used in battery, which is interesting and might attract lots of attention to this hot topic. The reviewer supports the publication. Though this review is a comprehensive document, if some more discussion could go to the disadvantages of the self-healing materials, would be better. The future is bright, while the difficulties to achieve should also make it clear.
Reply: Thank you very much for your positive comments. Due to the adjustment of the content and structure of the manuscript, we have rewritten Chapter 2 (“The mechanisms of self-healing materials”, from page 2 to page 5) to make it more concise in terms of language and content. We reclassified the self-healing mechanisms by deleting the classification of extrinsic and intrinsic methods and unifying them into physical, chemical and physico-chemical approaches. Self-healing materials used in lithium-ion batteries are basically based on chemical approaches, and physical approaches are barely used, so we only discussed about chemical approaches.
In Chapter 2, we have mentioned the disadvantages of self-healing materials based on chemical approaches: “However, since the inherent dynamic chemical responses of the SH materials are usually slow, this self-healing mechanism dictates that the healing process may take longer” (on page 3, line 92). In section 2.1, we have also mentioned the disadvantages of self-healing materials based on supramolecular interactions: “Another attractive feature of supramolecular chemistry is that the matrix can be re-paired rapidly and autonomously, but the mechanical strength of the repaired sites is usually much lower compared to covalent and free-radical rebonding” (on page 3, line 115). The disadvantages have also been discussed in other parts of the manuscript, such as the section 7.1 “Challenges”: “Compared with other self-healing materials, intrinsic self-healing materials have advantages in volumetric and gravimetric energy density, but the slow response and self-healing process limit their commercialization. How to improve the self-healing efficiency is a major challenge in future research” (on page 23, line 881).
In a word, we not only discussed about the advantages, but also summarized the disadvantages, and all the revisions are in “Track Changes” mode and have been high lightened in the manuscript.
Reviewer 3 Report (New Reviewer)
The subject dealt with by the authors of this work, Self-Healable Lithium-Ion Batteries, is very interesting and of great relevance for the progress in the improvement of batteries, whose demand is really important nowadays.
However, the distribution of the different sections of the review, which contains a large number of subsections, does not generate an easy reading, being repetitive on many occasions. As an example, the phrase “As mentioned above” is replicated up to five times in the manuscript and as the first sentence in multiple paragraphs.
There are too many levels of sections, many of them with almost no developed content, that sometimes resulting in a short paragraph by section. It is suggested to abbreviate and group the theme. There are examples to be grouped:
3. 2 Self-healing binders
3.2.1. Supramolecular interactions
3.2.1.1 Hydrogen bonding
3.2.1.2. Metal-ligand coordination
3.2.1.3. Host-guest interactions
3.2.1.4. Dynamic ionic bonding
3.2.2. Dynamic covalent bonds
3.2.3. Dual crosslinking
3.2.4. Summary
4. Self-healing electrolytes
4.1. Self-healing gel polymer electrolytes
4.1.1. Liquid electrolytes
4.1.2. Ionic liquids
4.2 Self-healing all-solid-state electrolytes
4.2.1 Solid polymer electrolytes
4.2.2 Solid composite electrolytes
6 Self-healing interfaces
6.1 Self-healing electrolyte-electrode interfaces
6.2.1 Based on liquid metal and alloys
6.2.2 Based on supramolecular interactions
7. Challenges and opportunities
7.1. Challenges
7.2. Opportunities
above, "Summary" could be included as a final paragraph instead of a subsection. Likewise, “Challenges and Conclusions” may be another paragraph, which should be rewritten because is repeated and inconclusive.
In conclusion, the distribution with a large number of subsections makes it difficult to read.
Review the typographical error (there are several). Review the figures:
- Figure 1 should be changed. Its circular distribution makes it difficult to read with the letters backwards. This circular structure is not appropriate to do the outline the outline of the review.
- Incorrect Figure caption of Fig2. Specify Fig. 2e and 2f and translate (h) in this figure.
Author Response
Referee: 3
Comments to the Author:
The subject dealt with by the authors of this work, Self-Healable Lithium-Ion Batteries, is very interesting and of great relevance for the progress in the improvement of batteries, whose demand is really important nowadays.
However, the distribution of the different sections of the review, which contains a large number of subsections, does not generate an easy reading, being repetitive on many occasions. As an example, the phrase “As mentioned above” is replicated up to five times in the manuscript and as the first sentence in multiple paragraphs.
There are too many levels of sections, many of them with almost no developed content, that sometimes resulting in a short paragraph by section. It is suggested to abbreviate and group the theme. There are examples to be grouped:
- 2 Self-healing binders
3.2.1. Supramolecular interactions
3.2.1.1 Hydrogen bonding
3.2.1.2. Metal-ligand coordination
3.2.1.3. Host-guest interactions
3.2.1.4. Dynamic ionic bonding
3.2.2. Dynamic covalent bonds
3.2.3. Dual crosslinking
3.2.4. Summary
- Self-healing electrolytes
4.1. Self-healing gel polymer electrolytes
4.1.1. Liquid electrolytes
4.1.2. Ionic liquids
4.2 Self-healing all-solid-state electrolytes
4.2.1 Solid polymer electrolytes
4.2.2 Solid composite electrolytes
6 Self-healing interfaces
6.1 Self-healing electrolyte-electrode interfaces
6.2.1 Based on liquid metal and alloys
6.2.2 Based on supramolecular interactions
- Challenges and opportunities
7.1. Challenges
7.2. Opportunities
above, "Summary" could be included as a final paragraph instead of a subsection. Likewise, “Challenges and Conclusions” may be another paragraph, which should be rewritten because is repeated and inconclusive.
In conclusion, the distribution with a large number of subsections makes it difficult to read.
Review the typographical error (there are several). Review the figures:
- Figure 1 should be changed. Its circular distribution makes it difficult to read with the letters backwards. This circular structure is not appropriate to do the outline the outline of the review.
- Incorrect Figure caption of Fig2. Specify Fig. 2e and 2f and translate (h) in this figure.
Reply: Thank you very much for your positive suggestions. We have carefully revised this manuscript according to your valuable suggestions.
- The distribution of the different sections of the review, which contains a large number of subsections, does not generate an easy reading, being repetitive on many occasions. As an example, the phrase “As mentioned above” is replicated up to five times in the manuscript and as the first sentence in multiple paragraphs.
There are too many levels of sections, many of them with almost no developed content, that sometimes resulting in a short paragraph by section. It is suggested to abbreviate and group the theme.
Reply: Thank you very much for your positive comments and we have carefully revised this draft according to your valuable suggestions. Firstly, we have rewritten Chapter 2 (“The mechanisms of self-healing materials”, from page 2 to page 5) to make it more concise in terms of language and contents. Secondly, we deleted unnecessary subheadings (i.e., 3.2.1.1 Hydrogen bonding, 3.2.1.2. Metal-ligand coordination, 3.2.1.3. Host-guest interactions, 3.2.1.4. Dynamic ionic bonding) to make the article easy to read. Thirdly, we have reduced the number of repetitive phrases, such as “As mentioned above”. All the revisions are in “Track Changes” mode and have been high lightened in the manuscript.
- 2. "Summary" could be included as a final paragraph instead of a subsection.
Reply: Thank you very much for your positive comments. We have included all the subsection “Summary” parts as the final paragraph instead of a subsection to make the manuscript more concise and easier to read.
- Likewise, “Challenges and Conclusions” may be another paragraph, which should be rewritten because is repeated and inconclusive.
Reply: Thank you very much for your positive comments. Although some of the chapters (i.e., Chapter 5. Self-healing current collectors) are short, we believe that it is necessary to keep them independent and cannot be recombined. As well, we also keep some of the necessary subheadings (i.e., 4.2.1 Solid polymer electrolytes, 4.2.2 Solid composite electrolytes) to make the manuscript logically clear.
- 4. Review the typographical error (there are several). Review the figures:
- Figure 1 should be changed. Its circular distribution makes it difficult to read with the letters backwards. This circular structure is not appropriate to do the outline the outline of the review.
- Incorrect Figure caption of Fig2. Specify Fig. 2e and 2f and translate (h) in this figure.
Reply: Thank you very much for your positive comments. We have fixed the grammatical errors in the article and all the revisions are in “Track Changes” mode and have been high lightened in the manuscript. We have also re-edited figure 1 and deleted the original figure 2, figure 3 and figure 4. Actually, we have rewritten Chapter 2 (“The mechanisms of self-healing materials”, from page 2 to page 5) to make it more concise in terms of language and content. This is because we realized that some kinds of self-healing materials are not used in lithium-ion batteries, and their mechanisms also have no relationship with lithium-ion batteries. In order to highlight the main point of this article, we reclassified the self-healing mechanisms by deleting the classification of extrinsic and intrinsic methods and unifying them into physical, chemical and physico-chemical approaches. Since self-healing materials used in lithium-ion batteries are basically based on chemical approaches, and physical approaches are barely used, we only discussed about chemical approaches. Therefore, as a schematic, figure 1 has been changed from the original one focusing on self-healing mechanisms to the new one focusing on the applications of self-healing materials in lithium-ion batteries. The original figure 2, figure 3 and figure 4 have been deleted and the new figure 2 mainly exhibited the mechanisms of chemical approaches which are related to the self-healing materials in lithium-ion batteries. All the changes have been made to make the article more focused, concise, and easier to read. All the revisions are in “Track Changes” mode and have been high lightened in the manuscript.
Round 2
Reviewer 3 Report (New Reviewer)
Accept in present form
This manuscript is a resubmission of an earlier submission. The following is a list of the peer review reports and author responses from that submission.
Round 1
Reviewer 1 Report
The subject of this review is to make an assessment of self-healing materials, in a first step the different types of self-healing materials are presented, this part is not directly related to batteries and should be shorter.
The article is relatively well structured, the main part is associated with the stabilisation of the negative electrodes, only a few families of electrodes are described, in particular silicon electrodes, no study on lithium metal are included, which should be revised. The selected articles are largely described, but are little commented. This review includes a set of information, capacity, number of cycles, without specifying the improvement obtained due to the presence of self-healing materials and more damagingly without critical contribution. In particular the chemical and electrochemical stabilities of binders and others are never discussed. No comment is also made on the cost of the proposed proposals, and their feasibility. The figures are often of very poor quality, and the quality has to be improve.
Before being published, this review needs to be extensively revised, in particular by highlighting the contribution or otherwise of the self-healing materials, and by discussing in depth the other properties of the materials.
Reviewer 2 Report
Given the persistent safety concerns regarding high energy batteries approaches to improved safety are very much welcome. This quite obviously includes self-healing options. This very relevant topic was unfortunately badly handled by the present authors. The English style need to be revised. They have prepared the manuscript with a lot of mistakes in writing, highlighted by the reviewer. The manuscript needs to be highly revised.
